# Robust Feature-Level Adversaries are Interpretability Tools

**Stephen Casper**[*123], **Max Nadeau**[*234], **Dylan Hadfield-Menell**[1], **Gabriel Kreiman**[23]

[1]MIT CSAIL; [2] Boston Children's Hospital, Harvard Medical School;
[3]Center for Brains, Minds, and Machines; [4]Harvard College, Harvard University
scasper@mit.edu     mnadeau@college.harvard.edu
* Equal Contribution

## Abstract

The literature on adversarial attacks in computer vision typically focuses on pixel-level perturbations. These tend to be very difficult to interpret. Recent work that manipulates the latent representations of image generators to create "feature-level" adversarial perturbations gives us an opportunity to explore perceptible, interpretable adversarial attacks. We make three contributions. First, we observe that feature-level attacks provide useful classes of inputs for studying representations in models. Second, we show that these adversaries are uniquely versatile and highly robust. We demonstrate that they can be used to produce targeted, universal, disguised, physically-realizable, and black-box attacks at the ImageNet scale. Third, we show how these adversarial images can be used as a practical interpretability tool for identifying bugs in networks. We use these adversaries to make predictions about spurious associations between features and classes which we then test by designing "copy/paste" attacks in which one natural image is pasted into another to cause a targeted misclassification. Our results suggest that feature-level attacks are a promising approach for rigorous interpretability research. They support the design of tools to better understand what a model has learned and diagnose brittle feature associations.[1]

## 1   Introduction

State-of-the-art neural networks are vulnerable to adversarial examples. Conventionally, adversarial inputs for visual classifiers take the form of small-norm perturbations to natural images [66, 18]. These perturbations reliably cause confident misclassifications. However, to a human, they typically appear as random or mildly-textured noise. Consequently, it is difficult to interpret these attacks–they rarely generalize to produce human-comprehensible insights about the target network. In other words, beyond the observation that such attacks are possible, it is hard to learn much about the underlying target network from these pixel-level perturbations.

In contrast, many real-world failures of biological vision are caused by perceptible, human-describable features. For instance, the ringlet butterfly's predators are stunned by adversarial "eyespots" on its wings (Appendix A.1, Fig. 7). This falls outside the scope of conventional adversarial examples because the misclassification results from a feature-level change to an object/image. The adversarial eyespots are robust in the sense that the same attack works across a variety of different observers, backgrounds, and viewing conditions. Furthermore, because the attack relies on high-level features, it is easy for a human to describe it.

---

[1]https://github.com/thestephencasper/feature_level_adv

36th Conference on Neural Information Processing Systems (NeurIPS 2022).

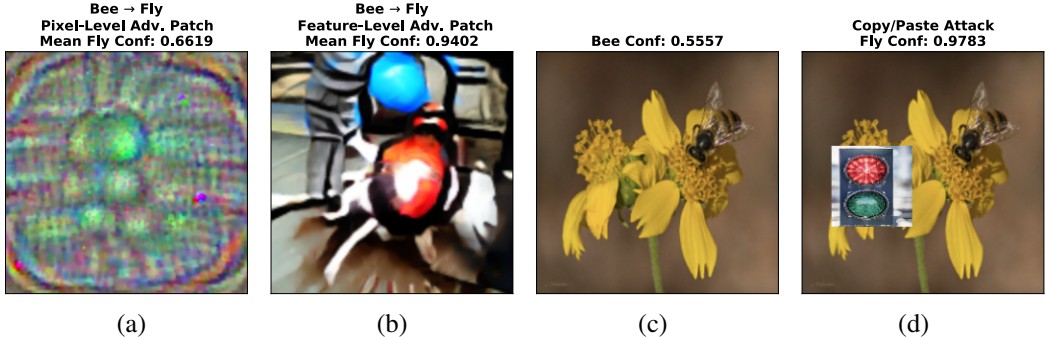

Figure 1: Our feature-level adversaries are useful for interpreting deep networks (we used a ResNet50 [21]). (a) A pixel-level adversarial patch trained to make images of bees misclassified as flies. (b) An analogous feature-level adversarial patch. (c) A correctly-classified image of a bee. (d) A successful copy/paste attack whose design was guided by adversarial examples like the one in (b).

This work takes inspiration from the ringlet butterfly's eyespots and similar examples in which a model is fooled in the real world by an interpretable feature (e.g. [46]). Our goal is to design adversaries that reveal easily-understandable weaknesses of the victim network. We focus on two desiderata for adversarial perturbations: attacks must be (1) interpretable (i.e. describable) to a human, and (2) robust so that interpretations generalize. We refer to these types of attacks as "feature-level" adversarial examples. Several previous works have created attacks by perturbing the latent representations of an image generator (e.g., [24]), but thus far, approaches have been small in scale, limited in robustness, and not interpretability-driven (See Section 2).

We build on this prior work to propose an attack method that generates feature-level attacks against computer vision models. This method works on ImageNet scale models and creates robust, feature-level adversarial examples. We test three methods of introducing adversarial features into source images either by modifying the generator's latents and/or inserting a generated patch into natural images. In contrast to previous works that have enforced the "adversarialness" of attacks only by inserting small features or restricting the distance between an adversary and a benign input, we also introduce methods that regularize the feature to be perceptible yet disguised to resemble something other than the target class.

We show that our method produces robust attacks that provide actionable insights into a network's learned representations. Fig. 1 demonstrates the interpretability benefits of this type of feature-level attack. It compares a conventional, pixel-level, adversarial patch, created using the method from [6], with a feature-level attack using our method. While both attacks attempt to make a network misclassify a bee as a fly, the pixel-level attack exhibits high-frequency patterns and lacks visually-coherent objects. On the other hand, the feature-level attacks displays easily describable features: the colored circles. We can validate this insight by considering the network performance when a picture of a traffic light is inserted into the image a bee. In this example, the image classification moves from a 55% confidence that the image is a bee to a 97% confidence that the image is of a fly. Section. 4.2 studies these types of "copy/paste" attacks more in depth.

Our contributions are threefold.

1. **Conceptual Insight:** We observe that robust feature-level adversaries can used to produce useful types of inputs for studying the representations of deep networks

2. **Robust Attacks:** We introduce methods for generating feature-level adversaries that are uniquely versatile and able to produce targeted, universal, disguised, physically-realizable, and black-box attacks at the ImageNet scale. See Table 1.

3. **Interpretability:** We generalize from our adversarial examples to design copy/paste attacks, verifying that our adversaries help us understand the network well enough to exploit it.

The following sections contain background, methods, experiments, and discussion. Appendix A.11 has a high-level summary for a lay-audience. Code is available at https://github.com/thestephencasper/feature_level_adv.

| | Targeted | Universal | Disguised | Physically-Realizable | Transferable/Black-Box | Copy/Paste | ImageNet Scale |
|---|---|---|---|---|---|---|---|
| Szegedy et al. (2013) [66], Goodfellow et al. (2014) [18] | ✔ | ✗ | ✗ | ✗ | ✗ | ✗ | ✔ |
| Natural mimics, e.g. Peacock, Ringlet Butterfly | ✔ | ✔ | ✗ | ✔ | ✔ | ✗ | N/A |
| Hayes et al. (2018)[20] | ✔ | ✔ | ✗ | ✗ | ✔ | ✗ | ✔ |
| Mopuri et al. (2018)a[41] | ✔ | ✔ | ✗ | ✗ | ✔ | ✗ | ✔ |
| Mopuri et al. (2018)b[42] | ✔ | ✔ | ✗ | ✗ | ✔ | ✗ | ✔ |
| Poursaeed et al. (2018) [51] | ✔ | ✔ | ✗ | ✗ | ✔ | ✗ | ✔ |
| Xiao et al. (2018) [73] | ✔ | ✗ | ✗ | ✗ | ✔ | ✗ | ✔ |
| Hashemi et al. (2020) [19] | ✔ | ✔ | ✗ | ✗ | ✔ | ✗ | ✔ |
| Wong et al. (2020) [72] | ✔ | ✗ | ✗ | ✗ | ✗ | ✗ | ✗ |
| Liu et al. (2018) [38] | ✔ | ✗ | ✔ | ✗ | ✗ | ✗ | ✗ |
| Samangouei et al. (2018) [55] | ✔ | ✗ | ✗ | ✗ | ✗ | ✗ | ✗ |
| Song et al. (2018) [63] | ✔ | ✗ | ✔ | ✗ | ✔ | ✗ | ✗ |
| Joshi et al. (2018) [29] | ✔ | ✗ | ✗ | ✗ | ✗ | ✗ | ✗ |
| Joshi et al. (2019) [28] | ✔ | ✗ | ✔ | ✗ | ✗ | ✗ | ✗ |
| Singla et al. (2019) [60] | ✔ | ✗ | ✗ | ✗ | ✔ | ✗ | ✗ |
| Hu et al. (2021) [24] | ✔ | ✔ | ✔ | ✔ | ✗ | ✗ | ✗ |
| Wang et al. (2020) [69] | ✔ | ✔ | ✔ | ✗ | ✗ | ✗ | ✗ |
| Kurakin et al. (2016) [34] | ✔ | ✗ | ✗ | ✔ | ✔ | ✗ | ✔ |
| Sharif et al. (2016) [57] | ✔ | ✗ | ✗ | ✔ | ✔ | ✗ | ✔ |
| Brown et al. (2017) [6] | ✔ | ✔ | ✗ | ✔ | ✔ | ✗ | ✔ |
| Eykholt et al. (2018) [15] | ✔ | ✗ | ✔ | ✔ | ✗ | ✗ | ✔ |
| Athalye et al. (2018) [2] | ✔ | ✗ | ✗ | ✔ | ✗ | ✗ | ✔ |
| Liu et al. (2019) [37] | ✔ | ✗ | ✗ | ✔ | ✔ | ✗ | ✔ |
| Thys et al. (2019) [67] | ✔ | ✔ | ✗ | ✔ | ✗ | ✗ | ✗ |
| Kong et al. (2020) [32] | ✔ | ✗ | ✗ | ✔ | ✗ | ✗ | ✗ |
| Komkov et al. (2021) [31] | ✔ | ✔ | ✗ | ✔ | ✗ | ✗ | ✗ |
| Dong et al. (2017) [11] | ✔ | ✗ | ✗ | ✗ | ✔ | ✗ | ✔ |
| Geirhos et al. (2018) [17] | ✗ | ✗ | ✗ | ✗ | ✗ | ✗ | ✔ |
| Leclerc et al. (2021) [35] | ✗ | ✗ | ✔ | ✗ | ✗ | ✔ | ✔ |
| Wiles et al. (2022) [70] | ✗ | ✗ | ✔ | ✗ | ✔ | ✗ | ✔ |
| Carter et al. (2019) [7] | ✔ | ✗ | ✔ | ✗ | ✗ | ✔ | ✔ |
| Mu et al. (2020) [43] | ✗ | ✗ | ✔ | ✗ | ✗ | ✔ | ✔ |
| Hernandez et al. (2022) [23] | ✗ | ✗ | ✔ | ✗ | ✗ | ✔ | ✔ |
| **Ours** | ✔ | ✔ | ✔ | ✔ | ✔ | ✔ | ✔ |

Table 1: Our feature-level attacks are uniquely versatile. Each row represents a related work (in the order in which they are presented in Section 2.) Each column indicates a demonstrated capability of a method. Note that two methods each having a ✔ for a capability does not imply they do so equally well. *Targeted*=working for an arbitrary target class. *Universal*=working for any source example. *Disguised*=Perceptible and resembling something other than the target class. *Physically-realizable*=working in the physical world. *Transferable/black-box*=transferring to other classifiers. *Copy/Paste*=useful for designing attacks in which a natural feature is pasted into a natural image.

## 2 Related Work

Here, we contextualize our approach with others related to improving on conventional adversarial examples [66, 18]. Table 1 summarizes capabilities.

**Inspiration from Nature:** Mimicry is common in nature, and sometimes, rather than holistically imitating another species, a mimic will only display particular features. For example, many animals use adversarial eyespots to confuse predators [64] (see Appendix A.1 Fig. 7a). Another example is the mimic octopus which imitates the patterning, but not the shape, of a banded sea snake. We show in Figure 7b that a ResNet50 classifies an image of one as a sea snake.

**Generative Modeling:** An approach related to ours has been to train a generator or autoencoder to produce small adversarial perturbations that are applied to natural inputs. This has been done to synthesize imperceptible attacks that are transferable, universal, or efficient to produce [20, 41, 42, 51, 73, 19, 72]. Rather than training a generator, ours and other works have perturbed the latents of pretrained generative models to produce perceptible alterations. [38] did this with a differentiable image renderer. Others [55, 63, 29, 28, 60, 24] have used deep generative networks, and [69] aimed to create more semantically-understandable attacks by using an autoencoder with a "disentangled" embedding space. Our work is different in four ways. (1) These works focus on small classifiers trained on simple datasets (MNIST [36], Fashion MNIST [74], SVHN [44], CelebA [39], BDD [75], INRIA [9], and MPII [1]) while we work at the ImageNet [53] scale. (2) We do not simply rely on

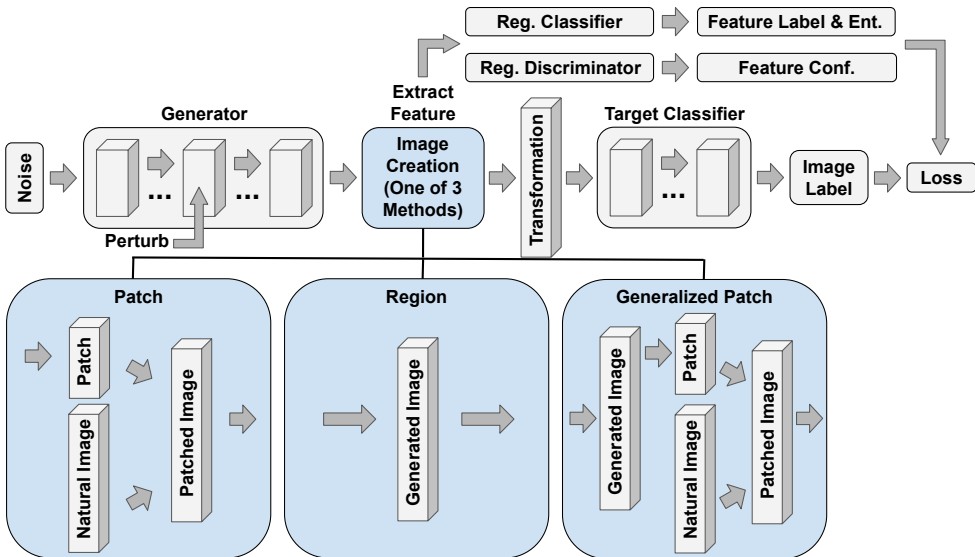

Figure 2: Our fully-differentiable pipeline for creating feature-level attacks. In each experiment, we create either "patch," "region," or "generalized patch" attacks. The regularization terms in the loss based on an external classifier and discriminator are optional and are meant to make the inserted feature appear disguised as some non-target class.

using small features or restricting the distance to a benign image to enforce the adversarialness of attacks. We introduce techniques that regularize the adversarial feature to be perceptible yet disguised to resemble something other than the target class. (3) We evaluate three distinct ways of inserting adversarial features into images. (4) Our work is interpretability-oriented.

**Attacks in the Physical World:** Physical-realizability demonstrates robustness. We show that our attacks work when printed and photographed. This directly relates to [34] who found that pixel-space adversaries could do this to a limited extent in controlled settings. More recently, [57, 6, 15, 2, 37, 67, 32, 31] created adversarial clothing, stickers, or objects. In contrast with these, we also produce attacks in the physical world that are disguised as a non-target class.

**Adversaries and Interpretability:** Using adversarial examples to better interpret networks has been proposed by [11] and [68]. We use ours to discover human-describable feature/class associations learned by a network. This relates to [17, 35, 70] who debug networks by searching over transformations, textural changes, and feature feature alterations. More similar to our work are [7, 43, 23], who use feature visualization [47] and network dissection [3] to interpret the network. Each use their interpretations to design "copy/paste" attacks in which one natural image pasted inside another causes an unrelated misclassification. We add to this work with a new method to identify such adversarial features. Unlike any previous approach, ours does so in a way that allows for targeted attacks that take into account an arbitrary distribution of source images.

## 3 Methods

We adopt the "unrestricted" adversary paradigm [63] under which an attack is successful if the network's classification differs from an oracle's (e.g., a human). Our adversaries can only change a small, fixed portion of either the generator's latent or the image. We use white-box access to the network, though we present black-box attacks based on transfer from an ensemble in Appendix A.6.

Our attacks involve perturbing the latent representation in some layer of an image generator to produce an adversarial feature-level alteration. Fig. 2 depicts our approach. We test three types of attacks, "patch", "region" and "generalized patch" (plus a fourth in Appendix A.5 which we call "channel" attacks). We find patch attacks to generally be the most successful.

**Patch:** We use the generator to produce a square patch that is inserted into a natural image [58].

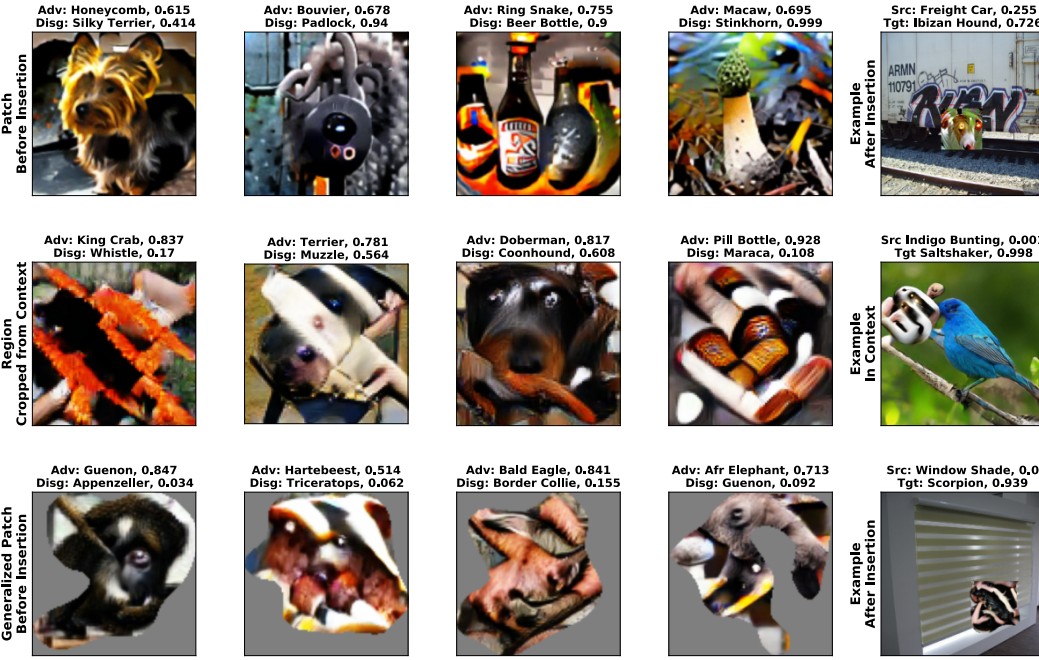

Figure 3: Examples of targeted, universal feature-level adversaries from patch (top), region (middle), and generalized patch (bottom) attacks. The first four columns show the adversarial features. The mean target class confidence is labeled 'Adv.' and is calculated under random source images (and random insertion locations for patch and generalized patch attacks). The target network's disguise class confidence for each patch or extracted generalized patch is labeled 'Disg.' The final column shows examples of the features applied to images. The example image for each is labeled with its source and target class confidences.

**Region:** Starting with some generated image, we randomly select a square column of the latent in a generator layer which spans the channel dimension and replace it with a learned insertion. This is analogous to a square patch in the pixel representation. We keep insertion location fixed over training. The modified latent is passed through the rest of the generator, producing the adversarial image.

**Generalized Patch:** These patches can be of any shape, hence the name "generalized" patch. We first generate a region attack and then extract a generalized patch from it. We do this by taking the absolute-valued pixel-level difference between the original and adversarial image, applying a Gaussian filter for smoothing, and creating a binary mask from the top decile of these pixel differences. We apply this mask to the generated image to isolate the region that the perturbation altered. We can then treat this as a patch and overlay it onto an image in any location.

**Basic Attacks:** For all attacks, we train a perturbation $\delta$ to the latent of the generator to minimize a loss that optimizes for both attacking the classifier and appearing interpretable:

$$\arg\min_{\delta} \mathbb{E}_{x\sim\mathcal{X},t\sim\mathcal{T},l\sim\mathcal{L}} \quad L_{\text{x-ent}}[C(A(x,\delta,t,l)),y_{\text{targ}}] + L_{\text{reg}}[A(x,\delta,t,l)] \qquad (1)$$

with $\mathcal{X}$ a distribution over source images (e.g., a dataset or generation distribution), $\mathcal{T}$ a distribution over transformations, $\mathcal{L}$ a distribution over insertion locations (this only applies for patches and generalized patches), $C$ the target classifier, $A$ an image-generating function, $L_{\text{x-ent}}$ a targeted crossentropy loss for attacking the classifier, $y_{\text{targ}}$ the target class, and $L_{\text{reg}}$ a regularization loss. The adversary has no control over $\mathcal{X}$, $\mathcal{T}$, or $\mathcal{L}$, so it must learn features that work on the network independent of any particular source image, transformation, or insertion location. For all of our attacks, $L_{\text{reg}}$ contains a total variation loss, $TV(a)$, to discourage high-frequency patterns.

**"Disguised" Attacks:** Ideally, a feature-level adversarial example should appear to a human as easily-describable but should not resemble the attack's target class. We call such attacks "disguised." Here, the main goal is not to fool a human, but to help them *learn* about what types of realistic features might cause the model to make a mistake. To train these disguised attacks, we use additional

terms in $L_{\text{reg}}$ as proxies for these two criteria. We differentiably resize the patch or the extracted generalized patch and pass it through a GAN discriminator and auxiliary classifier. We then add weighted terms to the regularization loss based on the discriminator's ($D$) logistic loss for classifying the input as fake, the output entropy ($H$) of some classifier ($C'$), and/or the negative of the classifier's crossentropy loss for labeling the input as the attack's target class. Note that $C'$ could either be the same or different than the target classifier $C$. With all of these terms, the regularization objective is

$$L_{\text{reg}}(a) = \lambda_1 TV(a) + \underbrace{\lambda_2 L_{\text{logistic}}[D(P(a))] + \lambda_3 H[C'(P(a))] - \lambda_4 L_{\text{x-ent}}[C'(P(a), y_{\text{targ}})]}_{\text{``Disguise'' Regularizers}}. \quad (2)$$

Here, $P(a)$ returns the extracted and resized patch from adversarial image $a$. In order, these three new terms encourage the adversarial feature to (1) look realistic, and (2) look like some specific class, but (3) not the target class. The choice of disguise class is left entirely to the training process.

## 4 Experiments

We use BigGAN generators from [5, 71], and perturb the post-ReLU outputs of the internal 'GenBlocks.' We also found that training slight perturbations to the BigGAN's inputs improved performance. We used the BigGAN discriminator and adversarially trained classifiers from [13] for disguise regularization. By default, we attacked a ResNet50 [21], restricting patch attacks to 1/16 of the image and region and generalized patch attacks to 1/8. Appendix A.2 has additional details. First, in Section 4.1 we show that these feature level adversaries are highly robust to suggest that interpretations based on them are generalizable. Second, in Section 4.2 we put these interpretations to the test and show that our feature level adversaries can help one understand a network well enough to exploit it.

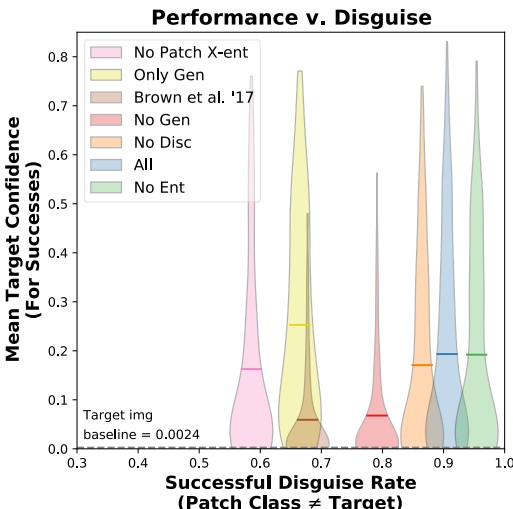

### 4.1 Robust Attacks

Figure 3 shows examples of targeted, universal, and disguised feature-level patch (top), region (middle), and generalized patch (bottom) attacks which were each trained with all of the disguise regularization terms from Eq. 2. We find the disguises to be effective, particularly for the patches (top row), but imperfect. Appendix A.3 discusses this and what it may suggest about networks and size bias.

**Performance versus Disguise:** Here, we study our patch attacks in depth to test how effective they are at attacking the network and how successfully they can help to identify non-target-class features that can fool the network. We compared seven different approaches. The first was our full approach using the generator and

Figure 4: Targeted, universal patch attacks compared. Successful disguise success rate (x axis) shows the proportion of attacks in which the patch was not classified by the network as the target class when viewed on its own. Mean target class confidence (y axis) gives the empirical target class confidences of 250 patch attacks. Each is an average over 100 source images. The proportion of each distribution above 0.5 gives a lower-bound for the top-1 attack success rate. The mean target class confidence for using randomly-sampled natural target class images as patches is 0.0024 and is shown as a thin dotted line at the bottom.

all disguise regularization terms from Eq. 2. The rest were ablation tests in which we omitted the generator (No Gen), the discriminator (No Disc) regularization term (No Reg), the entropy regularization term (No Ent), the crossentropy regularization term (No Patch X-ent), all three regularization terms (Only Gen), and finally the discriminator and all three regularization terms (Brown et al. '17). This final unregularized, pixel-level method resulted in the same approach as Brown et al. (2017) [6]. For each test, all else was kept identical including penalizing total variation, training under transformations, and initializing the patch as a generator output.

| Backpack: 0.0537 | Banana: 0.0009 | Jean: 0.0049 | Bath Towel: 0.0016 | Sunglasses: 0.0051 |
| Ibizan Hound: 0.8213 | Schipperke: 0.9828 | Puffer: 0.9893 | Loggerhead: 0.9606 | Macaw: 0.9831 |

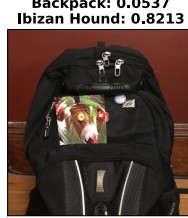 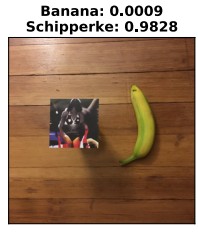 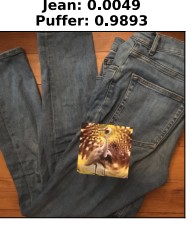 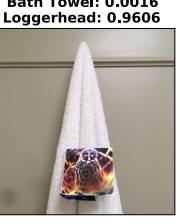 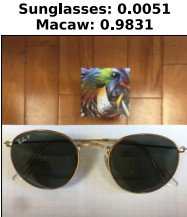

Figure 5: Examples of targeted, disguised, universal, and physically-realizable feature-level attacks. See Appendix A.10 Fig. 13 for full-sized versions of the patches.

For each method, we generated universal attacks with random target classes until we obtained 250 successfully "disguised" ones in which the resulting adversarial feature was not classified by the network as the target class when viewed on its own. Fig. 4 plots the success rate versus the distribution of target class mean confidences for each type of attack. For all methods, these universal attacks have variable target class confidences due in large part to the random selection of target class. Some attacks are stochastically Pareto dominated by others. For example, the pixel-space Brown et al. (2017) attacks were the least effective at attacking the target network and had the third least disguise rate. In other cases, there is a tradeoff between attack performance and disguise which can be controlled using the regularization terms from Eq. 2. We also compare our attacks to two baselines using resized natural images from the target class and randomly sampled patches from the center of target class images. These resulted in a mean target class confidences of 0.0024 and 0.0018 respectively.

Notably, Fig. 4 does not capture everything that one might care about in these attacks. It does not show any measure of how "realistic" the resulting patches look. In Appendix A.4, Fig. 9 plots the same target class confidence data from the $y$ axis in Fig. 4 versus the disguise class label confidence from an Inception-v3 which we use as a proxy for how realistic a human would find the patch. It suggests that the best attacks for producing patches that appear realistic are the "All" and "No Disc" methods. In Appendix A.10, Figs. 13, 14, and 15 give examples of successful "All", "Only Gen", and "Brown et al. '17" attacks respectively. Because they were initialized from generator outputs, some of the "Brown et al. '17" attacks have a veneer-like resemblance to non-target class features. Nonetheless, they contain higher-frequency patterns and less coherent objects in comparison to the two sets of feature-level attacks. We subjectively find the "All" attacks to be the best disguised.

**Physical-Realizability:** To test their ability to transfer to the physical world, we generated 100 additional targeted, universal, and disguised adversarial patches. We used the generator and all regularization terms (the "All" condition from above). We selected the 10 with the best mean target class confidence, printed them, and photographed each next to 9 objects from different ImageNet classes.[2] We confirmed that photographs of each object were correctly classified without a patch. Figure 5 shows successful examples. Meanwhile, resizable and printable versions of all these patches and others are in Appendix A.10. The mean and standard deviation of the target class confidences for our attacks in the physical world were 0.312 and 0.318 respectively ($n = 90$, not i.i.d.). This means that these patches' mean effectiveness dropped by less than $\frac{1}{2}$ when transferring to the physical world.

**Black-Box Attacks:** In Appendix A.6, we show that our targeted universal attacks can transfer from an ensemble to a held-out model.

### 4.2 Interpretability

If an adversarial feature successfully fools the victim network, this suggests that the network associates that feature in context of a source image with the target class. We find that our adversaries can suggest both beneficial and harmful feature-class associations. In Appendix A.7, Fig. 11 provides a simple example of each.

Simply developing an interpretation, however, is easy. Showing that one leads to a useful understanding of the network is harder. One challenge in the explainable AI literature is to develop interpretations that go beyond seeming-plausible and stand up to scrutiny [52]. Robust feature level adversarial patches can easily be used to develop hypotheses about the network's behavior, e.g. "The network

---

[2]Backpack, banana, bath towel, lemon, jeans, spatula, sunglasses, toilet tissue, and toaster.

thinks that bee features plus colorful balls implies a fly." But are these valid, useful interpretations of the network? In other words, are our adversaries adversarial because of their interpretable qualities, or is it because of hidden motifs? We verify interpretations by using our attacks to make and validate predictions about how to fool the target network with natural objects.

**Validating Interpretations with Copy/Paste Attacks:** A "copy-paste" attack is created by inserting one natural image into another to cause an unexpected misclassification. They are more restricted than patch attacks because the features pasted into an image must be natural objects. As a result, they are of high interest for physically-realizable attacks because they suggest combinations of real objects that yield unexpected classifications. They also have precedent in the real world. For example, subimage insertions into pornographic images have been used to evade NSFW content detectors [76].

To develop copy/paste attacks, we select a source and target class, generate class-universal adversarial features, and manually analyze them for motifs that resemble natural objects. Here, we used basic attacks without the disguise regularization terms from Eq. 2. We then paste images of these objects into natural images and pass them through the classifier.

Fig. 6 shows four types of copy/paste attacks. In each odd row, we show six patch, region, and generalized patch adversaries that were used to guide the design of a copy/paste attack. In each even row are the copy/paste adversaries for the 6 (of 50) images for the source class for which the insertion resulted in the highest target class confidence increase along with the mean target class confidences before and after patch insertion for those 6. The success of these attacks shows their usefulness for interpreting the target network because they require that a human understands the mistake the model is making like "Bee $\wedge$ Traffic Light $\rightarrow$ Fly" well enough to manually exploit it. Given the differences in the adversarial features that are produced in the Bee $\rightarrow$ Fly and Traffic Light $\rightarrow$ Fly attacks, Fig. 6 also demonstrates how our attacks take the distribution of source images into account.

**Comparisons to Other Methods:** Three prior works [7, 43, 23] have developed copy/paste attacks, also via interpretability tools. Unlike [43, 23], our approach allows for targeted attacks. And unlike all three, rather than simply identifying features associated with a class, our adversaries generate adversarial features for a target class *conditional* on any distribution over source images (i.e. the source class) with which the adversaries are trained. Little work has been done on copy/paste adversaries, and thus far, methods have either not allowed for targeted attacks or have required a human in the loop. This makes objective comparisons difficult. However we provide examples of a feature-visualization based method inspired by [7] in Appendix A.8 to compare with ours. For the Indian $\rightarrow$ African Elephant attack, the source and target class share many features, and we find no evidence that feature visualization is able to suggest useful features for copy/paste attacks. This suggests that our attacks' ability to take the source image distribution into account may be more helpful for discovering certain weaknesses compared to the baseline inspired by [7].

# 5 Discussion and Broader Impact

**Contributions:** Here we use feature-level adversarial examples to attack and interpret deep networks in order to contribute to a more practical understanding of network vulnerabilities. As an attack method, our approach is versatile. It can produce targeted, universal, disguised, physically-realizable, black-box, and copy/paste attacks at the ImageNet scale. This method can be also used as an interpretability tool to help diagnose flaws in models. We ground the notion of interpretability in the ability to make predictions about combinations of natural features that will make a model fail. And finally, we demonstrate this through the design of targeted copy/paste attacks for any distribution over source inputs.

**Implications:** Like any work on adversarial attacks, our approach could be used maliciously to make a system fail, but we emphasize their diagnostic value. Understanding threats is a prerequisite to avoiding them. Given the robustness and versatility of our attacks, we argue that they may be valuable for continued work to address threats that systems may face in practical applications. There are at least two ways in which these methods can be useful.

**Adversarial Training:** The first is for adversarial training. Training networks on adversarial images has been shown to improve their robustness to the attacks that are used [13]. But this does not guarantee robustness to other types of adversarial inputs (e.g. [22]). Our feature-level attacks are categorically different from conventional pixel-level ones, and our copy/paste attacks show

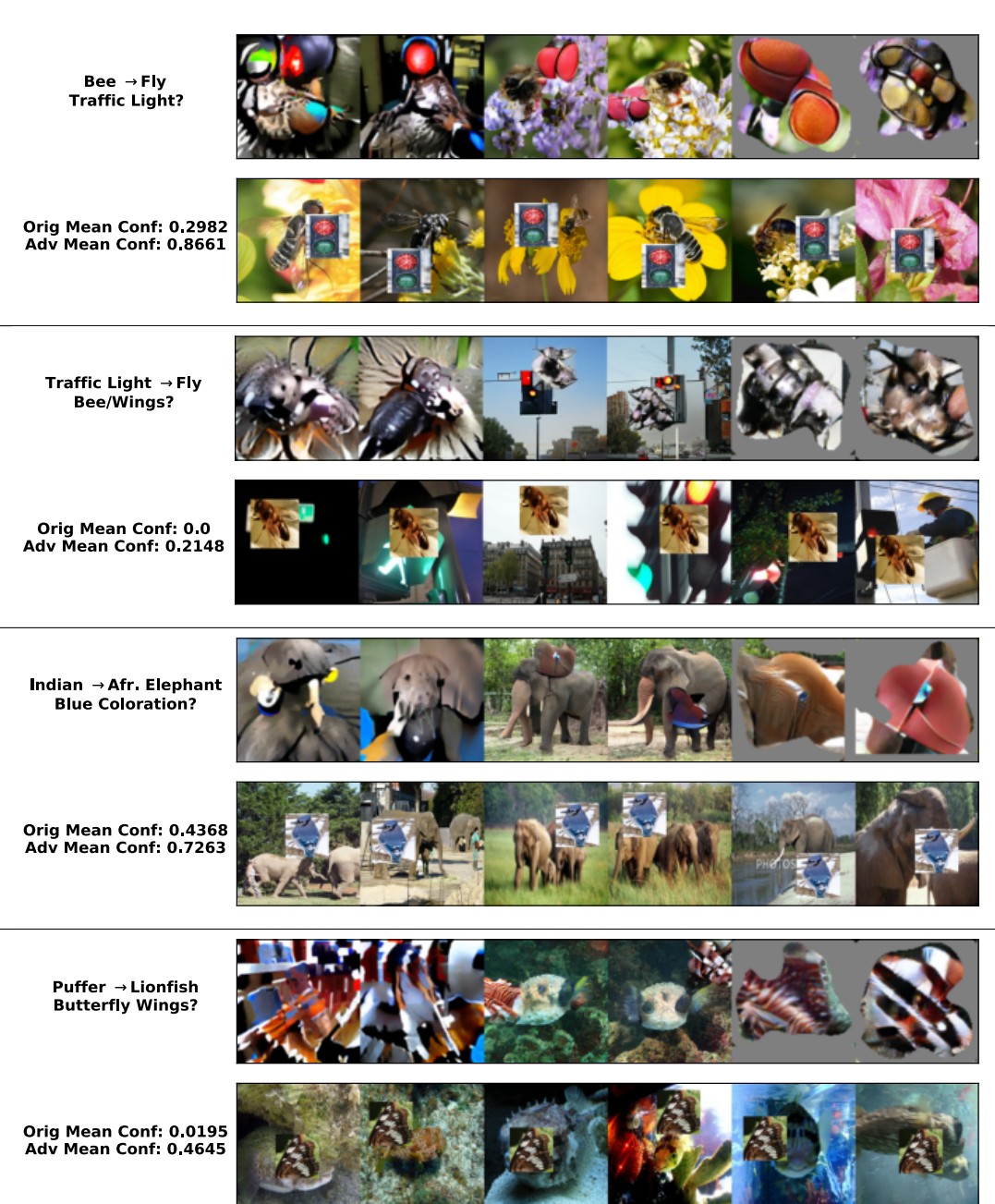

Figure 6: Feature-level adversaries can guide the design of class-universal copy/paste adversarial attacks. Patch adversary pairs are on the left, region in the middle, and generalized patch on the right of each odd row. Attack examples are on each even row. We do not claim that the traffic light ∧ bee → fly examples on row 4 are necessarily adversarial, but they demonstrate alongside the bee ∧ traffic light → fly adversaries that the adversarial features are sensitive to the the source images. For each attack except traffic light → fly, we limited ourselves to attempting 10 natural patches.

how networks can be fooled by novel combinations of natural objects, failures that are outside the conventional paradigm for adversarial robustness (e.g., [13]). Consequently, we expect that adversarial training on broader classes of attacks such as the one we propose here will be valuable for designing more robust models. As a promising sign, we show in Appendix A.9 that adversarial training is helpful against our attacks.

**Diagnostics:** The second is for rigorously diagnosing flaws. We show that feature-level adversaries aid the discovery of exploitable spurious feature/class associations (Fig. 6) and a socially-harmful bias (Appendix A.7 Fig. 11). Our approach could also be extended beyond what we have demonstrated here. For example, our methods may be useful for feature visualization [47] of a network's internal neurons. An analogous approach to ours can also be used in Natural Language Processing [62, 50], and we are currently working on a method for this. Furthermore, it may be valuable to use these adversaries to identify generalizable flaws in networks that humans can easily understand but with minimal human involvement. This would be much more scalable and prevent human priors from influencing interpretations. See [8] for follow up work involving the fully-automated discovery of copy/paste attacks.

**Limitations:** A limitation of our approach is that when multiple desiderata are optimized for at the same time (e.g., universality + transformation robustness + disguise), attacks are generally less successful, more time-consuming, and require more screening to find good ones. This could be a bottleneck for large-scale adversarial training. Ultimately, this type of attack is limited by the efficiency and quality of the generator, so future work should leverage advances in generative modeling. Our evaluation method is also limited to be a proof-of-concept for the design of copy/paste attacks. Future work should evaluate this more rigorously. We are currently working toward developing a benchmark for interpretability tools based on their ability to aid a human in rediscovering trojans [16] that have been implanted into a model.

**Conclusion:** As AI becomes increasingly capable, it becomes more important to design models that are reliable. Each of the 11 proposals for building safe AI outlined in [26] explicitly call for adversarial robustness and/or interpretability tools, and recent work from [78] on high-stakes reliability in AI found that interpretability tools strengthened their ability to produce inputs for adversarial training. Given the close relationship between interpretability and adversarial robustness, continued study of the connections between them will be key for building safer AI systems.

## Acknowledgments

We thank Cassidy Laidlaw, Miles Turpin, Will Xiao, and Alexander Davies for insightful discussions and feedback and Kaivu Hariharan for help with coding. This work was conducted in part with funding from the Harvard Undergraduate office of Research and Fellowships.

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
