

ResNet50 Output:

Sea Snake, 0.573
Night Snake, 0.052
King Snake, 0.042
Hognose Snake, 0.027
Sidewinder, 0.025

(a)                                                                    (b)

Figure 7: Examples of robust feature-level adversaries in nature. (a) A peacock and butterfly with adversarial "eyespots." (b) A 'mimic octopus' (image from [45]) mimics a banded sea snake's patterning and is classified as one by a ResNet50.

# A    Appendix

## A.1    Adversarial Features in Nature

Fig. 7 shows examples of robust feature-level adversaries in nature.

## A.2    Methodological Details

We found that performing crossentropy and entropy regularization (Eq. 2) for disguise using adversarially-trained auxiliary classifiers produced more easily interpretable results. This aligns with findings that adversarially-trained networks tend to learn more interpretable representations [14, 54] and better approximate the human visual system [56]. Thus, for crossentropy and entropy regularization, we used an $\epsilon = 4$ $L_2$ and $\epsilon = 3$ $L_\infty$ robust ResNet50s from [13]. For discriminator regularization, we use the BigGAN class-conditional discriminator with a uniform class vector input (as opposed to a one-hot vector). For patch adversaries, we train under colorjitter, Gaussian blur, Gaussian noise, random rotation, and random perspective transformations to simulate real-world variations in conditions. For region and generalized patch ones, we only use Gaussian blurring and horizontal flipping. Also for region and generalized patch adversaries, we promote subtlety by penalizing the difference from the original image using the LPIPs perceptual distance [77, 61]. All experiments were implemented with PyTorch [49], and implementations of all of our work can be run from Google Colab notebooks which we provide. We estimate that our project involved a total of < 400 hours of compute, mostly on 12GB NVIDIA Tesla K80 GPUs.

## A.3    How Successful are our Disguises?

All of our attacks are designed to be "adversarial" in either one or two different ways. First, all are limited to manipulating only a certain portion of the image or latent. Second, some of our attacks are trained to be disguised as discussed in Section 3. We subjectively find that our methods for disguise are generally useful . For example, compare Fig. 13 to Fig. 14. However, some of these features are not well disguised. Consider the patch of a at the top left of Fig. 13 whose target class is a pufferfish but which is disguised as a crane.

We find it clear that the patch depicts a crane, and the network agrees when shown the patch as a full image. However, the image also seems to have patterning and coloring that resembles a pufferfish. Furthermore, when shrunk to the size of a patch and inserted into a source image for an attack, the finer, more crane-like features may be less prominent, and the pufferfish-like ones may be more prominent. This suggests that the the adversary may be exploiting size biases.

Ultimately, to the extent that the patch resembles the target class, it is not disguised and arguably not adversarial. However, we emphasize three things. First, we find the disguise classes to generally be much more prominent in these patches than the target class. Second, being able to recognize target class features when the target class is known versus unknown are very different tasks. It is key to note such framing effects [27]. And third, very few attack methods are always successful at the ImageNet

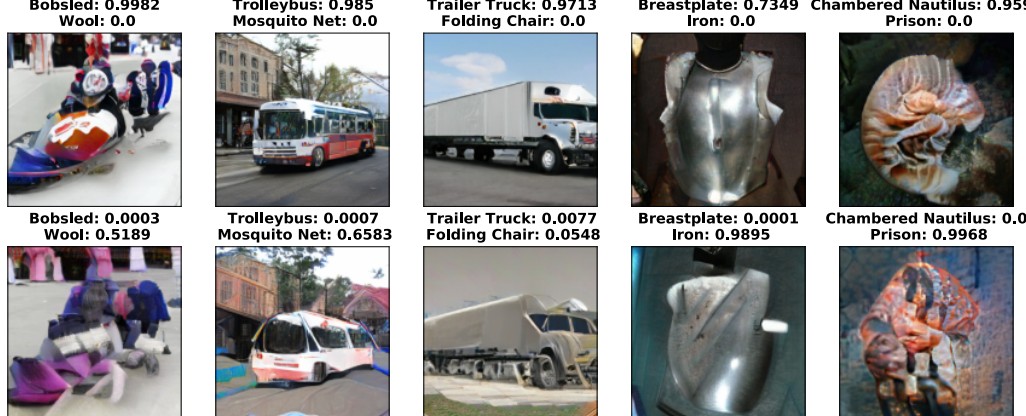

Figure 8: Examples of original images (top) alongside class-universal channel adversaries (bottom). Each image is labeled with the source and target class confidence.

scale, and some need for screening should be expected when optimizing an adversarial example for a complex objective (e.g. universality + transformation robustness + disguise).

We believe the studying human responses to feature-level adversaries and the links between interpretable representations, robustness, and similarity to human cortex [10] may be promising directions for better understanding both networks and biological brains.

### A.4  Attack Performance versus Disguise

In the main paper Section 4.1, Fig. 4 plots the successful disguise rate of attacks alongside their distribution of mean target class confidences. However, this leaves out how effective each type of attack is at appearing realistic to a human. Here, we use the target class confidence of an Inception-v3 [65] as a proxy for how realistic a patch appears to a human. Fig. 9 plots the mean target class confidences for successfully-disguised attacks versus their Inception-v3 disguise label confidence. This suggests that the attacks that are the best at producing realistic-looking patches are the "All" ones with the generator and all regularization terms and the "No Disc" ablations which omit the discriminator regularization term.

### A.5  Channel Attacks

In contrast to the region attacks presented in the main paper, we experiment here with *channel* attacks. For region attacks, we optimize an insertion to the latent activations of a generator's layer which spans the channel dimension but not the height and width. This is analogous to a patch attack in pixel-space. For channel attacks, we optimize an insertion which spans the height and width dimensions but only involves a certain proportion of the channels. This is analogous to an attack that only modifies the R, G, or B channel of an image in pixel-space. Unlike the attacks in Section 4, we found that it was difficult to create universal channel attacks (single-image attacks, however, were very easy). Instead, we relaxed this goal and created class-universal ones which are meant to cause any generated example from a source class to be misclassified as a target. We also manipulate $1/4^{th}$ of the latent instead of $1/8^{th}$ as we do for region attacks. Mean target class confidences and examples from the top 5 attacks out of 16 are shown in Fig. 8. They induce textural changes somewhat like adversaries crafted by [17] and [4].

### A.6  Black-Box Attacks

Adversaries are typically created using first order optimization on an input to a network which requires that the parameters are known. However, they are often transferrable between models [48], and one method for developing black-box attacks is to train against a different model and then transfer to the intended target. We do this for our adversarial patches and generalized patches by attacking a large ensemble of AlexNet [33], VGG19 [59], Inception-v3 [65], DenseNet121 [25], ViT[12, 40], and

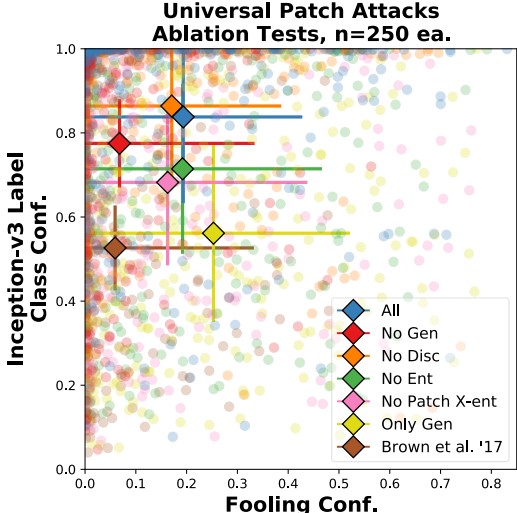

Figure 9: Targeted, universal patch attacks compared by mean target class confidence and Inception-v3 label-class confidence. Inception-v3 class conf. on the $x$-axis gives the mean target class confidence from the attacked network for images which have the patch inserted. The Inception-v3's label class confidence for the patch on the $y$-axis is used as a proxy for human interpretability. Attacks further up and right are better. Centroids are shown with error bars giving the standard deviation.

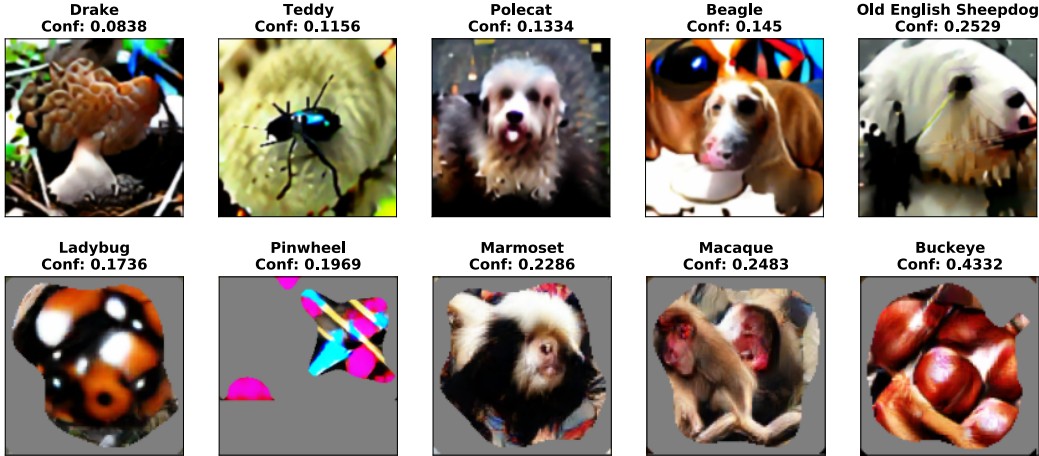

Figure 10: Universal black-box adversarial patches (top) and generalized patches (bottom) created using transfer from an ensemble. Patches are displayed alongside their target class and mean target class confidence.

two robust ResNet50s [13], and then transferring to ResNet50 [21]. Otherwise, these attacks were identical to the ones in Fig. 3 including a random source/target class and optimization for disguise. Many were unsuccessful, but a sizable fraction were able to work on the ResNet50 with a mean confidence of over 0.1 for randomly sampled images. The top 5 out of 64 of these attacks for patch and generalized patch adversaries are shown in Fig. 10.

## A.7    Discovering Feature-Class Associations

Fig. 11 shows two simple examples of using feature-level attacks to identify feature-class associations. It shows one positive example in which the barbershop class is desirably associated with barber-pole-stripe-like features and one negative example in which the bikini class is undesirably associated with

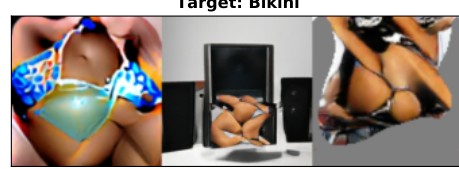

Figure 11: Examples of good and bad features class associations in which barber pole stripes are associated with a barbershop and a particular type of skin is associated with a bikini. Patch (left), region (middle), and generalized patch adversaries (right) are shown.

a particular type of skin. Notably the ability to simply identify feature associated with a target class is not a unique capability of our attacks and could also be achieved with feature visualization [47].

### A.8 Copy/Paste Attacks: Feature-Level Adversaries vs. Class Impressions

[43, 23, 7] each used interpretability methods to guide the development of copy/paste adversaries. [43] and [23], used network dissection [3] to develop interpretations of neurons and fit a semantic description to neurons which they combined with analysis of weight magnitudes. This allowed them to identify cases in which the networks learned undesirable feature-class associations. However, this approach cannot be used to make a targeted search for copy/paste attacks that will cause a given source class to be misclassified as a given target.

More similar to our work is [7] who found inspiration for successful copy/paste adversaries by creating a dataset of visual features and comparing the differences between ones which the network assigned the source versus target label. We take inspiration from this approach to create a baseline against which to compare our method for designing copy/paste attacks from Section 4.2. Given a source and target class such as a bee and a fly, we optimize a set of inputs to a network for each class in order to maximize the activation of the output node for that class. [42] refers to these as *class impressions*. We develop these inputs using the Lucent [30] package. We do this for the same source/target class pairs as in Fig. 6 and display 6 per class in Fig. 12. In each subfigure, the top row gives class impressions of the source class, and the bottom gives them for the target. Each class impression is labeled with the network's confidences for the source and target class. In analyzing these images, we find limited evidence of traffic-light-like features in the fly class impressions, and we find no evidence of more blue coloration in the African elephant class impressions than the Indian ones.

These class impressions seem comparable but nonredundant with our method from Section 4.2. However, our approach may have an advantage over the use of class impressions in that it is equipped to design features that look like the target class *conditional* on a distribution of source images. Contrastingly, a class impression is only meant to visualize features typical of the target class. This may be why our adversarial attacks are able to suggest that inserting a traffic light into a bee image or a blue object into an Indian elephant image can cause a misclassification as a fly or African elephant respectively. Bees and Flies and Indian and African elephant are pairs of similar classes. For cases in which the source and target are related, the class impressions may share many of the same features and thus be less effective for identifying adversarial features that work *conditional* on the source image distribution.

### A.9 Defense via Adversarial Training

Adversarial training is a common effective means for improving robustness. Here, to test how effective it is for our attacks, for 5 pairs of similar classes, we generate datasets of 1024 images evenly split between each class and between images with and without adversarial perturbations. This prevents the network from learning to make classifications based on the mere presence or absence of a patch. We do this separately for channel, region, and patch adversaries before treating the victim network as a binary classifier and training on the examples. We report the post-training minus pre-training accuracies in Tbl. 2 and find that across the class pairs and attack methods, the adversarial training improves binary classification accuracy by a mean of 42%.

|                          | Channel | Region | Patch | Mean |
|--------------------------|---------|--------|-------|------|
| **Great White/Grey Whale** | 0.49    | 0.29   | 0.38  | 0.39 |
| **Alligator/Crocodile**    | 0.13    | 0.29   | 0.60  | 0.34 |
| **Lion/Tiger**             | 0.29    | 0.28   | 0.63  | 0.40 |
| **Frying Pan/Wok**         | 0.32    | 0.39   | 0.68  | 0.47 |
| **Scuba Diver/Snorkel**    | 0.42    | 0.36   | 0.69  | 0.49 |
| **Mean**                   | 0.33    | 0.32   | 0.60  | **0.42** |

Table 2: Binary classification accuracy improvements from adversarial training for channel, region, and patch adversaries across 5 class pairs.

## A.10    Examples: Resizable, Printable Patches

See Figs. 13, 14, and 15 for feature-level and pixel-level control adversarial images. We encourage readers to experiment with these images (which were optimized to attack a ResNet50) or with others which can be created using our provided code. In doing so, one might find a mobile app to be convenient. We used Photo Classifier.[3]

## A.11    High-Level Summary

Here, we provide an easily-readable summary of this work, meant especially for readers who may be less familiar with machine learning jargon.

Historically, it has proven difficult to write conventional computer programs that classify images. But recently, immense progress has been made through neural networks which can now classify images into hundreds or thousands of categories with high accuracy. Despite this performance, we still don't fully understand the features that they use to classify images. Somewhat worryingly, past work has demonstrated that it is easy to take an image that the network classifies correctly and perturb its pixel values by a tiny amount in such a way that the network will misclassify it. For example, we can take a cat, make human-imperceptible changes to the pixels, and make the network believe that it is a dog. This process of designing an image that the network will misclassify is called an "adversarial attack."

Unfortunately, conventional adversarial attacks tend to produce perturbations that are not interpretable. To a human, they usually appear as pixelated noise (when exaggerated to be visible). As a result, they do not help us understand how networks will process human-describable inputs, and they do little to help us understand practical flaws in networks. Here, our goal is to develop different types of adversarial features that can be useful as debugging tools for these networks.

There are two properties that we want our adversarial images to have. First, we want them to be feature-level (not pixel-level) so that a human can interpret them, and second, we want them to be robust so that interpretations are generalizable. In one sense, this is not a new idea. Quite the opposite, in fact – there are examples of this the animal kingdom. Figure 7 shows examples of adversarial eyespots on a peacock and butterfly and adversarial patterns on a mimic octopus. These are interpretable, robust to a variety of viewing conditions, and give us useful information about how biological brains process these visual features.

The key technique that we use to create adversarial images involves an image generating network. Instead of perturbing pixels to cause an image to be misclassified, we perturb the internal state of the generator in order to induce a feature-level change to the output image. We also found that optimizing adversarial features under transformations to the images (like blurring, cropping, rotating, etc.) and using some additional terms in our optimization objective to encourage more realistic and better-disguised images improved results.

Our first key finding is that these attacks are very robust and versatile. For example, among other capabilities, we demonstrate that they can be universal (working for any image which we apply them to) and physically-realizable (working in the physical world when printed and photographed).

---

[3]https://apps.apple.com/us/app/photo-classifier/id1296922017

Our second key finding is that these adversaries are useful for interpreting networks. We use them as a way of producing useful classes of inputs for understanding ways that they can fail. Our strategy is to prove that these adversaries can help use understand the network well enough to exploit it. We analyze our adversarial features to get ideas for real-world features that resemble them. Then we test this interpretation by creating "copy/paste" attacks in which one natural image is pasted into another in order to cause a particular misclassification. Some of these are unexpected. For example, in Fig. 6, we find that a traffic light can make a bee look like a fly.

Together, our findings suggest that feature-level adversaries are very versatile attacks and practical debugging tools for finding flaws in networks. One implication is that by training networks on these adversaries, we might be able to make them more robust to failures that are due to feature-level properties of images. We also argue that these adversaries should be used as a practical debugging tool to diagnose problems in networks.

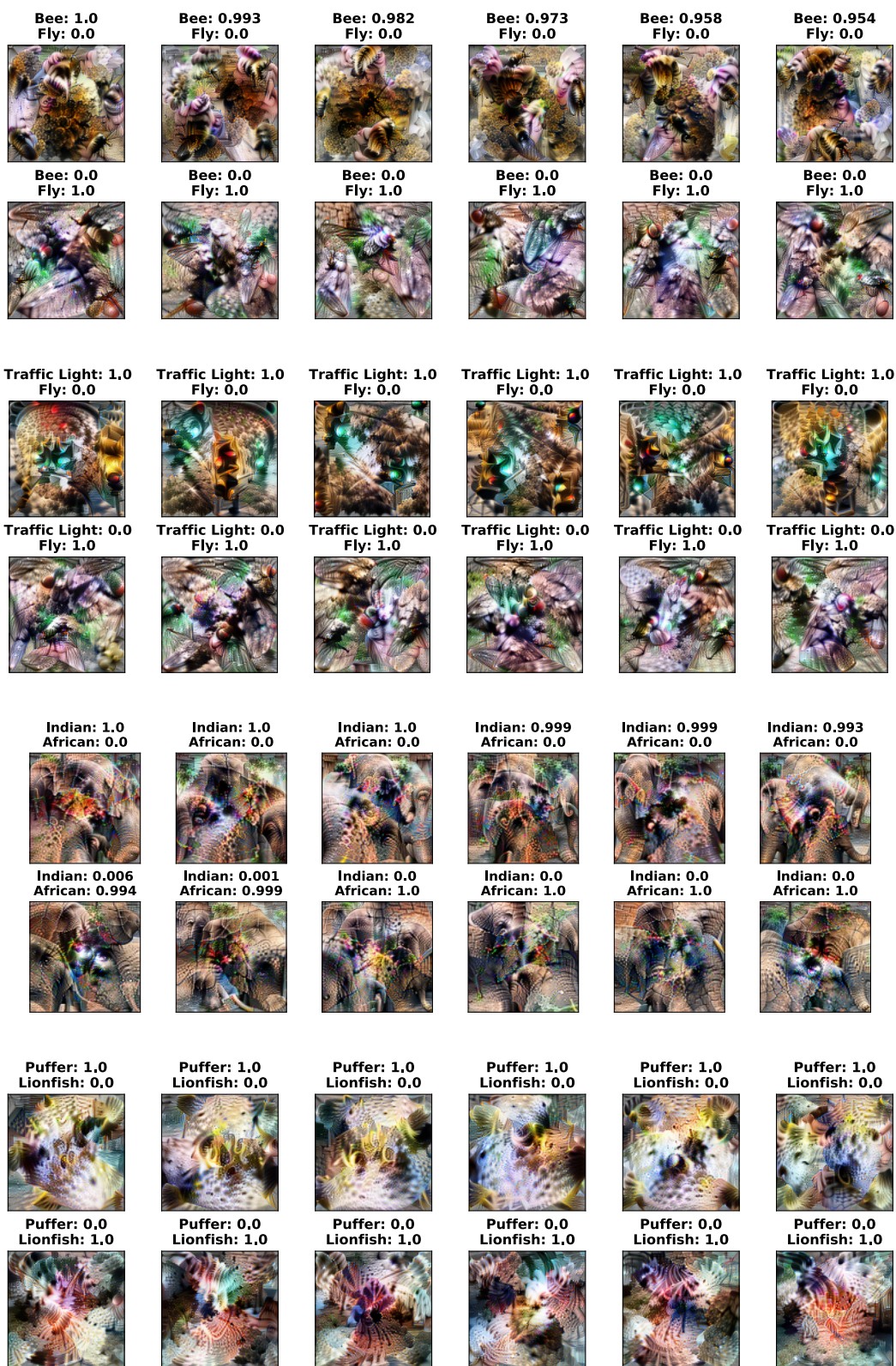

Figure 12: Class impressions for four pairs of classes. These could be used for providing insight about copy/paste attacks in the similar way to the examples from Fig. 6. Each subfigure is labeled with the network's output confidence for both classes.

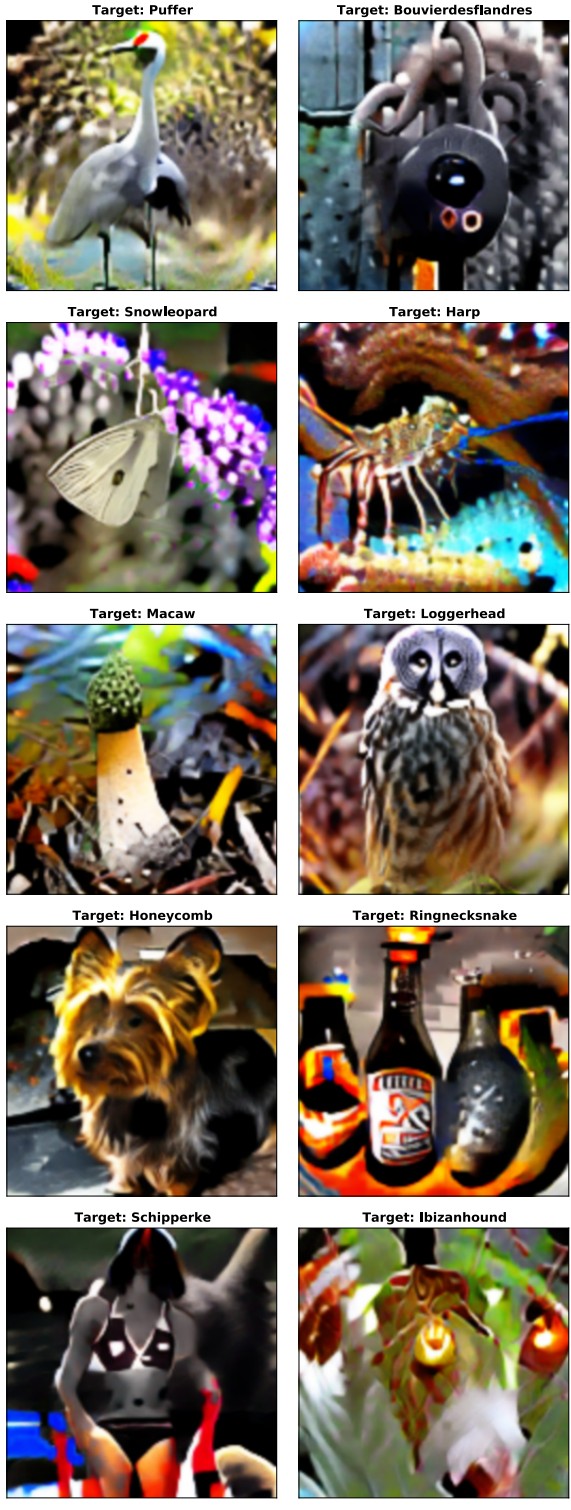

Figure 13: Examples of attacks created with a generator and with the disguise regularization terms from Eq. 2. See section 4.1 and the "All" datapoints from Fig. 4. We printed these to test physical realizability in Section 4.1

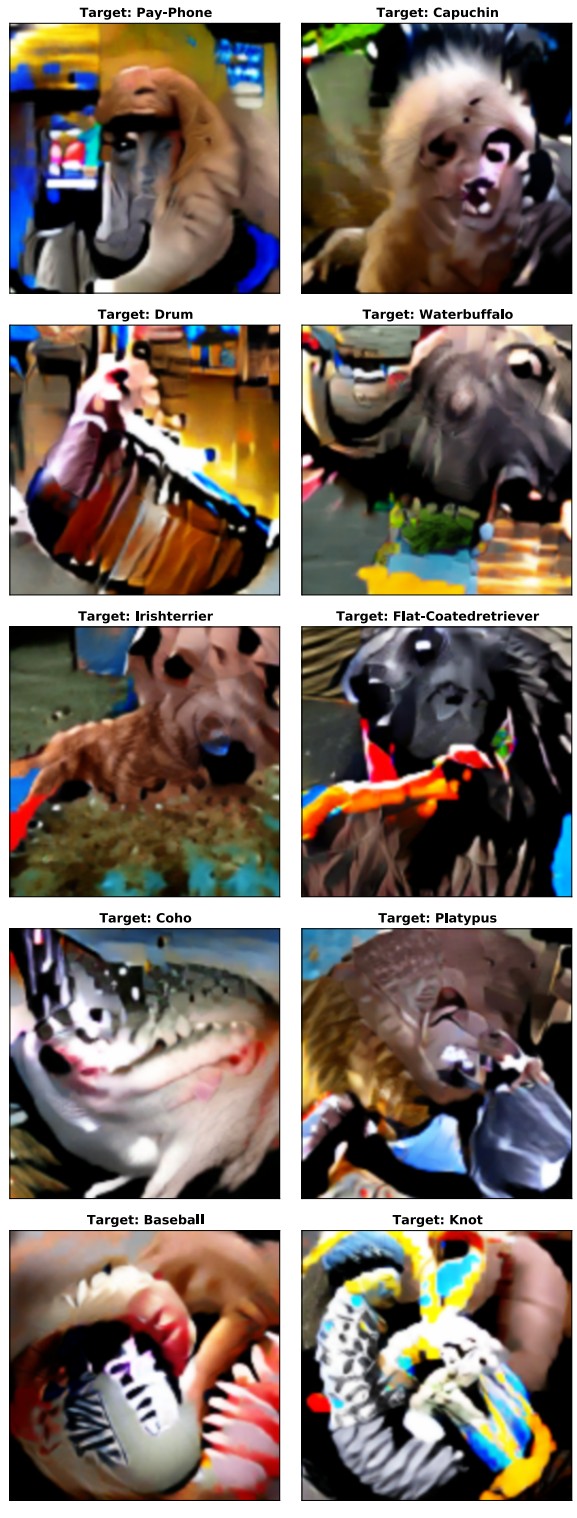

Figure 14: Examples of attacks created with a generator. See section 4.1 and the "Only Gen" datapoints from Fig. 4.

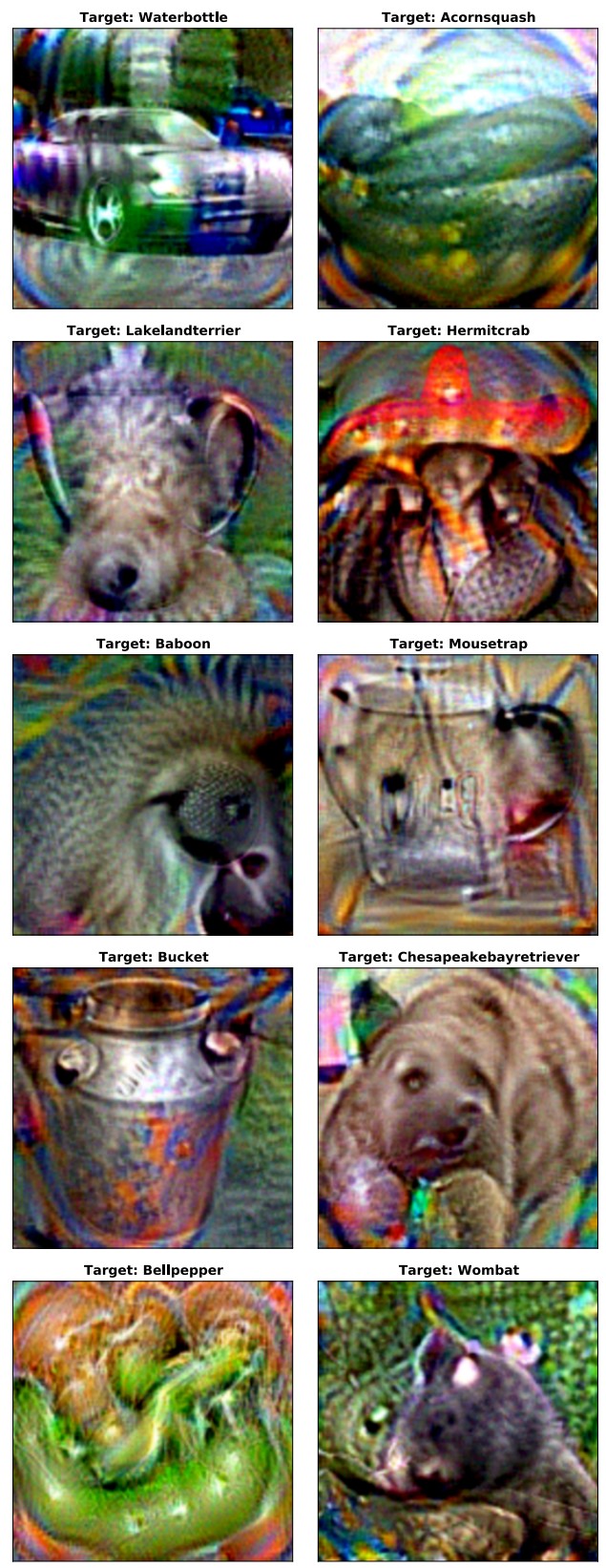

Figure 15: Examples of pixel-space attacks [6]. See section 4.1 and the "Brown et al. '17" datapoints from Fig. 4.