# OpenReview forum: "Robust Feature-Level Adversaries are Interpretability Tools"
_NeurIPS.cc/2022/Conference — NeurIPS 2022 Accept_

### Official Review · Reviewer_qpX2 · 2022-07-08

**Rating:** 7
**Confidence:** 3
**Soundness:** 4 excellent
**Presentation:** 2 fair
**Contribution:** 3 good

**Summary:**

The authors propose a new method for generating adversarial perturbations of images. These perturbations operate at a higher "feature-level" rather than at a pixel-level. As such, the perturbations are more understandable to humans and can potentially provide insights into model behavior.

**Questions:**

As mentioned in the "Weaknesses" section, I would strongly encourage the authors to use different wording when describing/framing their method.

**Limitations:**

The authors discussed the limitations of their work in a satisfactory manner.

**Strengths And Weaknesses:**

Strengths:

1. Through extensive experiments, the authors demonstrate that their method can provide insights on the behavior of deep-learning-based image classifiers. The perturbations produced by the authors' proposed method are indeed interpretable, and I found it quite interesting to see how the suggested perturbations changed model outputs in the provided examples. As such, I believe this method could be a valuable addition to the model-debugging toolbox to help machine learning practitioners better diagnose spurious correlations learned by their models.
2. To my knowledge (I am not an expert in the adversarial example literature), the method proposed by the authors is novel and the literature review seems very thorough.
3. I generally (with a couple exceptions, see Weaknesses section) found the writing to be high-quality and easy to follow.

Weaknesses:
1. My biggest issue with this work is not with the method itself, but rather with the vocabulary used to describe/frame it. Specifically:
    - To my knowledge (and indeed, this is in the second sentence of the paper), an "adversarial" attack involves modifying an image with a small-norm perturbation such that a deep learning model produces a wildly different output while a human does not realize the image has been modified. In this work the authors describe their method as a form of adversarial attack, even though the method is designed to produce perturbations that are obvious to human observers. For example, I wouldn't expect the "attacked" image in Figure 1d to be correctly classified after such a large perturbation. I would encourage the authors to find another way to frame their method (perhaps focusing more on the interpretability benefits?), as the current framing confused me when I first read the paper.
    - Similarly, I would highly recommend that the authors choose another word instead of "disguised" to describe something that is "perceptible and resembling something other than the target class". On my first reading, I assumed that "disguised" referred to something that is _imperceptible_.
2. The authors may wish to cite/discuss cycle-consistency-based methods for discovering human-interpretable perturbations that change the output of deep learning image classifiers (see e.g. [1] and an application in [2]).

[1]: "Explanation by Progressive Exaggeration" https://arxiv.org/abs/1911.00483
[2]: "AI for radiographic COVID-19 detection selects shortcuts over signal" https://www.nature.com/articles/s42256-021-00338-7

---

> ### Author Response · Authors · 2022-08-02
> **Response to qpX2**
>
> Thank you for the feedback, especially on clarity and terminology. We are glad that you find it a valuable addition to the model-debugging toolbox. Here are replies to points in order.
>
> **(1) Re: “To my knowledge (and indeed, this is in the second sentence of the paper), an "adversarial" attack involves modifying an image with a small-norm perturbation...For example, I wouldn't expect the "attacked" image in Figure 1d to be correctly classified after such a large perturbation. I would encourage the authors to find another way to frame their method...”**
>
> To reduce confusion, we modified the abstract and introduction to clarify that we are focusing on “perceptible” attacks. While most work on adversarial examples studies imperceptible attacks, we are not the first to study perceptible attacks generated by adversarial methods. For example, we see our characterization of adversarial attacks as similar to that of Brown et al., 2017 and past works on copy/paste attacks. All of these attacks can be seen as members of the class of unrestricted adversarial examples as Song et al, 2018 describe them: “images that successfully fool the classifier without confusing humans”.
>
> We agree that even random-images, if inserted as patches, could often make a network misclassify a source image. But we focus on _targeted_ misclassification, so in expectation over random source/target class selection and random patch selection, the mean target fooling rate for natural images as adversarial patches would be 1/num_classes or 0.001 for ImageNet classifiers. We agree that it feels demanding to ask Figure 1d to still be classified as a bee after modification. But classifying it as a fly (97% confidence) is clearly an error.
>
> On the other hand, we agree that the best baseline to compare against is not an unperturbed image. In response to this question, we added a baseline using natural images as patches to figure 4. Over 250 random natural images, we evaluated each as an “adversarial” patch for a random source/target pair. The mean fooling confidence for the target class was 0.0008. We added this baseline as a dotted line in Fig 4.
>
> **(2) Re: “Similarly, I would highly recommend that the authors choose another word instead of "disguised" to describe something that is "perceptible and resembling something other than the target class". On my first reading, I assumed that "disguised" referred to something that is imperceptible.”**
>
> Thank you for pointing this out. We have updated the paper to clearly state the definition of “disguised” upon its first use.
>
> **(3) Re: “The authors may wish to cite/discuss cycle-consistency-based methods.”**
>
> We thank the reviewer for pointing out this work. We added the “Explanation by Progressive Exaggeration” paper to related works and Table 1.

---

### Official Review · Reviewer_3cVn · 2022-07-10

**Rating:** 8
**Confidence:** 4
**Soundness:** 4 excellent
**Presentation:** 4 excellent
**Contribution:** 4 excellent

**Summary:**

The authors use a pre-trained generative model to automatically identify feature-level adversarial examples, which are adversarial perturbations that are both interpretable to a human and robust (i.e. still serve as adversarial attacks in related contexts). They target image classifier networks. To do this, they use several loss terms: A term that penalizes the predicted image class for being similar to the true image class; a term that penalizes high-frequency patterns; a term that encourages the realistic-ness of the image; a term that encourages the image to look like a particular class; and a term that encourages that class not to be the true class. They demonstrate the utility of all these loss terms with an ablation study. They evaluate three kinds of adversarial attack - (square) patch, region, and generalized (any shape) patch.  Impressively, they show that the attacks generated by their method are physically realizable and also transfer to a held out classifier. They are also able to use the attacks generated by the model to hand-craft copy/paste attacks, which is very useful for interpreting mistakes made by the target network.

**Questions:**

- How easy in general is it to construct copy/paste examples from adversarial examples generated by the method? The copy/paste attacks that were exhibited, were they cherry picked examples that worked? Or were they some of the few that the authors constructed and yet they worked?
- One suggestion for future research:  It is clearly be beneficial for interpretability to know where in the network an error is occurring. Perhaps the authors could regularize the features so that the adversarial features attack a particular layer of the target network.


**Limitations:**

The authors do an exceptionally good job of discussing the limitations and potential social impact of their work. They clearly make as much effort as can be expected of them to mitigate risks. Moreover they also engage constructively with a lay audience.

Although their work focuses on images, clearly there is much interest in the interpretability and robustness of non-image-based networks, such as language models. It isn't clear that gradient-based optimization (central to their method) will work in that domain due to the difficulty of generating realistic language examples from a generative model that uses an optimizable latent space, as is possible using image GANs. Given the considerable interest in the language domain, the authors might consider providing a slightly more detailed discussion than they currently do.


**Strengths And Weaknesses:**

The paper is very high quality, clearly presented, and probably high significance. Although the method itself makes from only minor alterations to previous work, what the paper lacks in originality, it more than makes up for in other factors including extensiveness of analyses, evaluation of societal impact, and provides a valuable tool to neural network interpretability.

A few criticisms of an otherwise great paper:

- It's often hard to identify the perturbation from the original image in e.g. fig 3. That hinders the reader's ability to understand the difference. I appreciate that an additional side-by-side comparison might make the figure unmanageably large, but I encourage the authors to figure out ways to make this key figure more useful to the reader.
- The description and labels in Fig 3 caption are unclear. "Each patch and generalized patch is labeled with its mean fooling confidence under random insertion in source images (labeled ‘Adv’) and the confidence for the disguise class (labeled ‘Img’)". Is 'disguise class' missing a -d, as in 'disguised class'? And the Img and Adv labels are confusing to me - it isn't perfectly obvious which is the class of the original and which is the predicted class after adversarial attack.
- The authors should make it even clearer what the difference between their work and Brown et al. (2017) is.  Is it simply that the present work uses a generator and Brown et al. do not?
- Figure 6 could be clearer. It isn't obvious at a glance which row ought to be paired with which other(s). Moving associated rows closer to each other would solve this.
- I feel like the 'bad' example in Fig 11 is perhaps a bit contrived: It's not obviously caucasian skin. Certainly not in the patch adversary. And caucasian would not have been my first guess for the skin colour of the region adversary or generalized patch adversary. I'd encourage the authors to find a more convincing example.

---

> ### Author Response · Authors · 2022-08-02
> **Response to 3cVn**
>
> Thank you for the feedback, especially on figures and the copy/paste attack methodology. We are glad that you find it high quality and high significance. Here are replies to points in order.
>
> **(1) Re: “It's often hard to identify the perturbation from the original image in e.g. fig 3…”**
>
> Thank you for highlighting this lack of clarity. We updated Fig. 3. It now shows universal adversarial features for all three methods out of any particular context and all with the same measures of success: mean fooling confidence, and disguise label class confidence.
>
> **(2) Re: “The description and labels in Fig 3 caption are unclear…”**
>
> For the patches and generalized patches, we do not display any information about how robust the victim network is to these attacks because we test patches on many source images. Instead, “Adv” referred to the target network’s average confidence in the target class.  “Img” refers to the network’s label class confidence when shown the patch in isolation. We changed “Img” to “Disg.” and updated the caption alongside the figure to make this clear.
>
> **(3) Re: “The authors should make it even clearer what the difference between their work and Brown et al. (2017) is. Is it simply that the present work uses a generator and Brown et al. do not?”**
>
> We apologize for the imprecise language. The difference between our full method and Brown et al.’ 2017’s (i.e. blue v. brown in Fig 4) is the generator and the three regularization terms introduced in Section 3. However, the use of total variation regularization and initialization of the image as an unperturbed generator output were the same. We updated Section 4.1 to state this explicitly.
>
> **(4) Re: “Figure 6 could be clearer. It isn't obvious at a glance which row ought to be paired with which other(s)...”**
>
> Thanks for the suggestion! We have added horizontal separation lines to make this more clear.
>
> **(5) Re: “I feel like the 'bad' example in Fig 11 is perhaps a bit contrived: It's not obviously caucasian skin…”**
>
> We agree. We updated the discussion to remove claims that it looks Caucasian. Instead, we simply state that it reveals certain shades of skin.
>
> **(6) Re: “How easy in general is it to construct copy/paste examples from adversarial examples generated by the method?”**
>
> Thank you for pointing this out. We agree that we should have been more clear about the approach we used to generate copy/paste attacks. We have updated Section 4.2 accordingly and summarize the method here. In short, we selected a few individual examples of images for copy/paste attacks, tested them as adversarial patches, and selected the best-performing ones. Unfortunately, we did not track time spent generating attacks. However, the process took minutes, not hours. For all examples except Traffic-Light to Fly, we checked fewer than 10 candidates images. We tested the  Traffic-Light to Fly attack with around 30 candidate images. We also note that previous work that considers copy/paste attacks (i.e.,  Carter et al., 2017 and Mu et al., 2021) do not report how they screened for examples.
>
> Furthermore, we acknowledge that our experiments are a proof of concept. This is a limitation of our work, which we intend to improve on in the future through a combination of 1) human subjects studies to evaluate the effort and ability of humans to design copy/paste attacks and 2) automated methods to design copy/paste attacks. We will update the future work section accordingly.
>
> **(7) Re: “It is clearly be beneficial for interpretability to know where in the network an error is occurring. Perhaps the authors could regularize the features so that the adversarial features attack a particular layer of the target network.”**
>
> Thank you for the suggestion. We agree that this would be useful, although we suspect that regularizing for parsimonious disruptions to neural activations may take some tuning and affect attack quality. If we were to try to understand what internal parts of the network were responsible for a vulnerability, the first thing we would probably try would be a tracing method using clean and adversarial examples to probe the network. See Meng et al., 2022 for a related example of tracing [ https://arxiv.org/abs/2202.05262 ]. We will add a discussion of this to future work.
>
> **(8) Re: “Although their work focuses on images, clearly there is much interest in the interpretability and robustness of non-image-based networks, such as language models…”**
>
> We agree that robust feature level adversaries would/will be interesting and probably very useful for studying language models. See Perez et al., 2022 [ https://arxiv.org/abs/2202.03286 ]. In fact, we are working on followup work that uses some analogous methods to identify and exploit interpretable failure modes in language models. As you mentioned, this process is less straightforward in NLP because a gradient cannot be taken through discrete text generation. We are adding this to our discussion of future work.

---

### Official Review · Reviewer_xUiw · 2022-07-11

**Rating:** 6
**Confidence:** 3
**Soundness:** 3 good
**Presentation:** 2 fair
**Contribution:** 2 fair

**Summary:**

The paper presents a method for generating feature-level adversarial examples. In the most successful variant of the attacks, adversarial perturbations in representation space of a generative model allow generation of “patches” which will then be added to natural images.The patches are optimized to be classified as “real” by a discriminator, increase entropy of a surrogate classifier, and misclassify the targeted classifier. The authors empirically verified the robustness of the adversarial examples, including its correctness in physical world. Then the authors study how the proposed method can be used for interpreting mistakes made by DNNs.

**Questions:**

1. The authors seem to be using a different objective for adversarial examples: they want the examples to be (1) easily-describable, (2) not classified as the target class, but have a rather loose restriction on how much perturbatitions can be made. The authors also agree their attacks may be detectable to humans. Thus to the best of my understanding, the contributions of this work is more on the interpretability side and not on the adversarial ML side, but the authors only talked about interpretability in one subsection--it would be great if the authors can elaborate more on how one could use the proposed method to diagnose the flaws.
2. Also related to the authors’ definition on disguised adversarial examples--it seems to me that natural images from any classes other than the targeted class also satisfy this definition, so a baseline to add would be use natural images from different classes to replace the adversarial patches. It would be nice to show why the proposed method is better than this baseline since it requires more computationally cost than the baseline.
3. Given the authors suggest this method can be used for interpretability, it would be good to test it on some language dataset since it would give some more intuitive outputs for understanding. This should be doable since there also exist generative language models.
It is worth to clarify what is the advantage of using feature-level adversarial examples for interpretability over other existing interpretability methods.
4. Some important parts of the paper is in the appendix, e.g. the black-box setting. It may be worth to reorganize the paper (e.g. in my mind most subfigures of figure 6 can be moved to the appendix).
The authors said the proposed method could be used for feature visualization. How would it be different from the existing feature visualization techniques?
5. It is not very intuitive to me why the surrogate classifier C’ is needed. Is it meant to make the adversarial examples transferable?
A minor typo: in figure 4, “Mean Target Confidence” is at the y-axis but in the caption it is called “fooling confidence”.


**Limitations:**

The authors stated the limitations of their work in the discussion and claimed that they are future works. Compared to other works on adversarial examples, the negative societal impact of this work is not significant since the generated adversarial examples here are easily detectable by human beings.

**Strengths And Weaknesses:**

Strength
The general method of the paper is presented relatively clear, and the idea of using feature-level adversarial examples for model interpretability.

Weakness
The focus of the paper is not clear: the proposed method has its advantage that it can be used as a interpretability tool, whereas as an attack the authors said it may be detectable by a human. However, the paper focus more on the attack than the interpretability. Besides, some important contents are placed in the appendix.

---

> ### Author Response · Authors · 2022-08-02
> **Response to xUiw**
>
> Thank you for the feedback. We appreciate the input. Here are replies to points in order.
>
> **(1) Re: “The focus of the paper is not clear...as an attack the authors said it may be detectable by a human. However, the paper focus more on the attack than the interpretability.”**
>
> We think the perceptibility of our adversaries is a feature, not a bug. It is what allows for our attacks to be used for interpretability.
>
> The results we want to emphasize the most are on interpretability, but (1) we want to highlight that our attacks are uniquely versatile compared to previous works. And (2) robustness is key to demonstrate to show our adversaries can be used to develop _generalizable_ interpretations of the network.
>
> Thank you for pointing out our contributions with interpretability seem underemphasized. We see the adversaries in general as the interpretability method and section 4.2 as the way we evaluate the usefulness of our interpretations. We have updated 4.2 to state that this is focused on evaluation of interpretations rather than interpretability.
>
> **(2) Re: “The authors..have a rather loose restriction on how much perturbatitions can be made. The authors also agree their attacks may be detectable to humans. Thus to the best of my understanding, the contributions of this work is more on the interpretability side ...but the authors only talked about interpretability in one subsection..”**
>
> We follow others, such as Brown et al., 2017 and Song et al, 2018, in calling detectable attacks “adversarial.” That being said, we see our main contribution to be on the interpretability side too, hence the title.
>
> Our evaluation of our interpretations goes _beyond_ the standards by which most works in the interpretability literature evaluate their methods. Few works test the ability of their methods to generate actionable insights for working with the network like we do with copy/paste attacks. We recently wrote a survey paper on interpretability methods that discusses the need for more rigorous evaluations. We have attached an anonymized version of our preprint in the supplemental material. Please see the “discussion” and “future works” sections of the survey for a discussion of the pervasiveness of poor evaluation.
>
> **(3) Re: “Also related to the authors’ definition on disguised adversarial examples--it seems to me that natural images from any classes other than the targeted class also satisfy this definition, so a baseline to add would be use natural images from different classes to replace the adversarial patches.”**
>
> Thank you for the suggestion.
>
> All of the attacks we present in the paper are _targeted_, so in expectation over random source/target class selection and random patch selection, the mean target fooling rate for natural images as adversarial patches would be 1/num_classes or 0.001 for ImageNet classifiers.
>
> On the other hand, we agree that the best baseline to compare against is not an unperturbed image. In response to you and 4JkH, we added a baseline from using natural images as patches to figure 4. We evaluated each as an “adversarial” patch for a random source/target class pair. The mean fooling confidence for the target class was 0.0008. We added this baseline as a dotted line in Fig 4.
>
> **(4) Re: “Given the authors suggest this method can be used for interpretability, it would be good to test it on some language dataset…”**
>
> We agree, and we are working on a followup paper to this one using language models as well! This work has some relations to Perez et al., 2022 [ https://arxiv.org/abs/2202.03286 ]. Things are more complicated in NLP because with modern language generators, outputs are produced by nondifferentiably sampling tokens. So some zero-order optimization technique like reinforcement learning must be used. We are excited about our work here, but we think that it should be be a stand-alone paper.
>
> **(5) Re: “The authors said the proposed method could be used for feature visualization. How would it be different from the existing feature visualization techniques?”**
>
> The closest paper of which we know to ours that focuses on feature visualization is Nguyen et al., 2016 [ https://arxiv.org/abs/1605.09304 ] which uses image generators. However, we know of no works that have used a regularization strategy for feature visualization like the one we present in Section 2. Just as our regularization terms can help us produce more realistic looking adversarial patches, we think they could be used to produce more realistic feature visualizations.
>
> **(6) Re: “It is not very intuitive to me why the surrogate classifier C’ is needed…”**
>
> Thank you for highlighting this issue in our presentation. In principle, the surrogate classifier could be the same as the victim (i.e., C’=C). We found empirically that using a different surrogate C’ (specifically, one that was trained on conventional adversarial examples) improved results. We have edited our explanation to make this more clear.

---

> > ### Comment · Reviewer_xUiw · 2022-08-06
> > **Thanks for your responses!**
> >
> >
> > 1+2: I agree with you that your focus is on interpretability, but that means you should have mentioned it early on in the paper, rather than in the last subsection before discussion
> >
> > 3: Same for their general response; why don't you consider the case that the inserted image is chosen from the target class?
> >
> > 4: Confirming that evaluation on language datasets will not be part of this current submission?
> >
> > 5: My problem here is that you are claiming a lot about the capabilities of the method, but provide limited empirical support.
> >
> > 6: Your response is not entirely convincing; more intuition is provided for why a surrogate model is needed.

---

> > > ### Author Response · Authors · 2022-08-07
> > > **Thanks for following up**
> > >
> > > 1+2. Thank you for this comment. We added some sentences to the beginning of section 4 to better signpost the experiments. After considering possibly ways to reorganize the paper, we do not feel that reordering the section or subsection structure of the paper would improve the flow. We also stress that our goal of interpretability is mentioned clearly in the title and abstract.
> > >
> > > 3. We conducted this experiment and will replace the baseline in figure 4 and the corresponding descriptions. Instead of a baseline of 0.0008, the new one is 0.0024.
> > >
> > > 4. Correct. We will not have NLP experiments in this paper.
> > >
> > > 5. We see this. We reworded the sentence mentioning this but made not other changes.
> > >
> > > 6. The only justification that we claim is that we tried both methods and found that using a different C' resulted in adversarial features that appeared to us to be more realistic. And we expected this given Engstrom et. al [1]. Unfortunately, we did not test the difference between normal and adversarially trained models in the figure 4 experiments.
> > >
> > > [1] Engstrom, L., Ilyas, A., Santurkar, S., Tsipras, D., Tran, B., & Madry, A. (2019). Adversarial robustness as a prior for learned representations. arXiv preprint arXiv:1906.00945.

---

### Official Review · Reviewer_4JkH · 2022-07-12

**Rating:** 4
**Confidence:** 4
**Soundness:** 2 fair
**Presentation:** 3 good
**Contribution:** 2 fair

**Summary:**

The paper proposes using feature-level adversarial perturbation to explore interpretable adversarial attacks. The proposed method can generate targeted, universal, disguised, physically realizable, and black-box attacks at the ImageNet scale. The method can also be used as an interpretability tool for detecting bugs in networks, and to evaluate this, the paper designs "copy/paste" attacks to cause targeted misclassification.

**Questions:**

- Figure 3 is confusing to me. Are all of the methods generating patches or localized perturbations? Why are patches (top) and generalized patches (bottom) shown as a complete picture instead of "a patch" (localized region) in a full image, while the region (middle) shown as a localized patch? I am asking because a patch will accompany by a "location" where the patch is located inside an image. This location will determine whether the patch requires to occlude the salient features of the attacked images, which will tell more about the attack (as shown in Figure 3 middle). In addition, because this paper is about "interpretability tools" as mentioned in the title, where the patch locates inside an image will also help to understand more about the interpretability of the network.

- Can the method control the location of the region attack, or whether the patch is sensitive to location? As shown in Figure 3 (middle), we can see that the patch seems to occlude the salient features to create the adversarial effect.

- How effective is the method compared with previous works? The paper seems to lack a quantitative comparison with previous methods. For example, to compare robustness, the paper should use a traditional metric of Attack Success Rate to measure how successful the patch can generalize and evade a large number of images to quantify the "universal" claim as well as to compare with other patch attacks. Similarly, for the "interpretability", how effective is the method compared to other interpretability tools mentioned in the literature in terms of interpreting networks? I recommend the paper conduct further experiments to clarify these points to quantify the superiority of the proposed method.

- What is the purpose of the "disguise" or "realistic" objective, and how to evaluate that better? I also wonder why the paper considers the "disguise" or "realistic" objective. I assume it is to fool human beings or at least to be realistic to a human. If that is the case, I suggest the paper use a better metric such as similarity metrics rather than simply relying on Inception-V3 or conduct a user study to evaluate this because it is hard to be convinced that the patches shown in Figures 3 and 4  are more realistic than previous patch methods.

- Minor questions:
	- In Fig 9. "Attacks further up and right are better?" ==> should it be further up and left?

**Limitations:**

I think the paper adequately addressed the limitations and potential negative societal impact.

**Strengths And Weaknesses:**

Strengths:
+ The paper is well written and easy to follow.
+ The method is evaluated on a large-scale ImageNet dataset.

Weaknesses:
- The evaluation of the "disguise" or "realistic" objective is not very convincing.
- The paper lacks a quantitative comparison with previous methods.
See the detailed questions in the next section.

---

> ### Author Response · Authors · 2022-08-02
> **Response to 4JkH**
>
> Thank you for the feedback.
>
> **(1) Re: “Figure 3 is confusing...”**
>
> The patch and generalized patch adversaries are optimized to fool the network from any location in any source image. For the region attacks, the adversary controls a randomly chosen but fixed part of the latent which is generated alongside the rest of the image; this is why we displayed them in the context of a background image.
>
> Thank you for pointing out the issue. We made a new version of Fig. 3 that shows region attacks out of context. And we now report for all three types of patches the same measures of success.
>
> **(2) Re: “...This location will determine whether the patch requires to occlude the salient features of the attacked images…Can the method control the location of the region attack...?...”**
>
> We apologize this was unclear. Per the above, adversaries do not control insertion location. We updated Section 3 to say this explicitly. We tested all adversaries on random source images and all patch and generalized patch adversaries under random insertion locations. Thus, our results are not affected by any adaptation to specific backgrounds. We updated Section 4 to say this explicitly.
>
> We agree occlusion matters. However, all experiments presented in the paper are _targeted_ attacks. So the expected  confidence in the target class for any occlusion-based technique is trivially upper bounded by 1/num_classes. To demonstrate this, we tested natural images as adversarial patches. The mean confidence in the target class was 0.0008. We added this to Fig 4.
>
> **(3) Re: “How effective is the method compared with previous works?...the paper should use a traditional metric of Attack Success Rate…“**
>
> Our controls encompass many previous works’ approaches (not just Brown et al., 2017) by testing the use of a generator, total variation regularization, and optimization under transformation. The only germane technique that we know of from a previous work that we did not use was a “non-printability” regularization term (Kurakin et al., 2016). But we omitted this because optimization under transformations (including colorjitter) served the same purpose.
>
> We used the mean confidence in the target class as our measure of performance. This is common in the literature. We do not present the top-1 fooling rate, but we can easily lower bound this with the proportion of the distributions in figure 4 over the 0.5 mark. We added this point to the paper.
>
> **(4) Re: “how effective is the method compared to other interpretability tools mentioned in the literature in terms of interpreting networks?”**
>
> To the best of our knowledge, the other papers that use interpretability tools for copy/paste attacks are Mu et al., 2020; Hernandez et al., 2022; and Carter et al. 2017. The methods used by the first two are based on identifying and exploiting polysemantic neurons. They cannot be used for _targeted_ copy/paste attacks, so we do not attempt a comparison.
>
> For Carter at al., 2017, we dedicate A.8 to comparing our methods. We do not attempt a quantitative one because both of these methods rely on a human in the loop. But this isn’t for a lack of wanting quantitative results. We have ongoing work to do this. We have added a discussion of this limitation and ongoing work to address it in the future work section of the paper.
>
> **(5) Re: “What is the purpose of the "disguise" or "realistic" objective, and how to evaluate that better?...I assume it is to fool human beings or at least to be realistic to a human.”**
>
> Disguise can be useful for guaranteeing “adversarialness” because if a feature resembles the target class, it’s not very adversarial.
>
> This is important but not our central concern. We have edited the paper to make it much clearer why we care about “disguised” and “realistic” attacks. The goal of them is not to fool the human. It is to help the human _learn_ about what features might cause the model to make a mistake.
>
> **(6) Re: “I suggest the paper use a better metric such as similarity metrics rather than simply relying on Inception-V3 or conduct a user study...”**
>
> We agree a user study would be best. However, (1) we stress that Inception-v3 is widely used in the image-generation literature to calculate “Inception Score” and “Frechet Inception Distance” which are proxies for how successful an image generator is [ https://link.springer.com/chapter/10.1007/978-3-030-63322-6_8 ]. We have updated figure 9 to clarify that the y axis is intended only as a proxy. And (2), while human trials would be more thorough, this type of experiment is ultimately not our focus. We are trying to provide interpretability tools for humans to use rather than images to fool humans.
>
> **(7) Re: “In Fig 9. "Attacks further up and right are better?" ==> should it be further up and left?”**
>
> Attacks further up and to the right are better. The X axis gives the confidence of the target model on the target class. We updated the caption to clarify this.

---

> > ### Comment · Reviewer_4JkH · 2022-08-09
> > **Thanks for your responses**
> >
> > (1) I think Figure 3 would be better if you show full images including the patches instead of only showing patches. This will help the readers understand how large is the patch in the context and whether they occlude the salient features of images.
> >
> > (2) Even for targeted attacks, the location of the patches is still important. For example, it is easier for the patch to fool the classifier/improve the confidence score if it occludes the salient features of source images because the patches are the main salient features now, while it would be more difficult if there are two or more salient features. We can observe these cases for all patches and examples in Figure 6.
> >
> > (3) Would it be much easier if the paper presents a plot regarding Attack Success Rate instead? A minor comment is that it would be much easier to check the revised version if the newly added texts are highlighted; for example by using a different color.
> >
> > (4) I agree it is hard to compare with a lack of quantitative metrics, the paper should clarify the main differences and highlight them in the paper.
> >
> > (5+6) If the main purpose of the paper is to "help the human learn about what features might cause the model to make a mistake" then I think the paper should explain or clarify further in detail how to interpret the results of your generated patches. In particular, what is the process to convert the results of patches to the copy/paste attacks in Figure 6? Is it human-dependent? Is it a trial-and-error process when guessing the images to conduct copy/paste attacks?  Are there any ways to make this process autonomous; for example, by comparing the results of different generated patches and providing a suggestion of a natural image for copy/paste attack? If it is human-dependent, then each human will have a different interpretability ability, would it be better if the interpretability of the generated patch to copy/paste attack is assessed by multiple different human beings? The paper should clarify these points to highlight the contribution of the paper.

---

> > > ### Author Response · Authors · 2022-08-09
> > > **Thank you for following up**
> > >
> > > (1) We have updated figure 3 again. It still has examples of the adversarial features as before. However we made the last column show the adversarial features applied to source images. We think that feedback on Fig. 3 from you and the other reviewers has been very helpful.
> > >
> > > (2) We agree about the importance of occlusion and its presence in Fig. 6. But we stress that occlusion by itself, while helpful for making the attacked image not appear as the source class, it not nearly as helpful for making the attacked image appear as a specific disguise class in general.
> > >
> > > (3) Our main motivation for reporting mean fooling confidence instead of top-1 fooling rate is that it can sometimes show evidence of success where top-1 error rate cannot. For example, in the traffic-light —> fly attacks in fig. 6, the bee patches on traffic-lights do not do a good job at top-1 fooling. But the mean fooling confidence shows that the bees do indeed make the images appear more fly-like — which was our goal in this case.
> > >
> > > We apologize that we did not highlight edits or include line numbers in the description of them in our earlier response.
> > >
> > > (4) Thank you. We made minor revisions our description of A.8’s experiments in the main paper to better state what we found. See lines 258-262.
> > >
> > > (5+6) In response to you and reviewer 3cVn, we added details of this process to the end of the caption of Fig. 6. In short, we manually searched for tested a small number of candidate natural patches. Ultimately though, these experiments are limited to being a proof of concept. As mentioned in our global response point 2(c) above, we are working on a future paper about removing human subjectivity in this process by making it fully-automated.

---

### Author Response · Authors · 2022-08-02
**Global Response from Authors**

Thank you to the reviewers for feedback and comments. We appreciate the input. We are glad to hear the comments that it was a “valuable addition to the model-debugging toolbox” (qpX2) and “high quality, clearly presented, and probably high significance” (3cVn). In addition to individual responses, here are a few general items we would like to respond to.

1. We apologize that the appendix was included in the main paper. This was a mistake, but we confirmed with an email to the area chairs on May 24 that this would not be an issue. The new submission fixes this.

2. There are several great ideas brought up in reviews that we are addressing, but with followup work. We have rewritten our discussion of future work to discuss these directly.

2a. 4JkH discusses a lack of _quantitative_ experiments to compare how useful our interpretability methods are to related ones. The dearth of quantitative methods to evaluate the utility of human-in-the-loop interpretability tools is a problem with the literature at large and is not unique to the current paper. But this is not an excuse. We agree and believe that the field needs better methods to quantify and compare different methods. We discuss this in an anonymized survey paper preprint of ours included in the new supplemental submission. And we are addressing this with followup work that evaluates the usefulness of different tools from the literature by their ability to help humans rediscover trojans that have been implanted into a network.

2b. Both xUiw and 3cVn noted that similar approaches may be interesting for language models. We agree, and are working on synthesizing interpretable examples that make large language models fail.

2c. 3cVn discussed the issue of quantifying specifically how much human effort went into the construction of copy-paste attacks. Effort is difficult to quantify, but we added the number of natural images we limited our attempts to in Section 4.2. And in a followup project, we are taking things a step further and introducing a method to develop copy/paste attacks using natural images and no human in the loop.

3. We have followed the reviewers’ suggestions and improved substantially on Figure 3, its caption, and the discussion of it. It now shows universal adversarial features for all three methods out of any particular context and all with the same measures of success: mean fooling confidence, and disguise label class confidence.

4. 4JkH, xUiw, and qpX2 each comment on how we do not account for the fact that simply inserting any images as patches into an image could fool the network. However, our attacks are targeted, so in expectation over randomly selected natural patches, and random source/target classes, an upper bound of the expected value for the mean confidence in the target class would be 1/num_classes (0.001 for ImageNet). On the other hand, we ran this experiment and added the results to Section 4.1 and figure 4. The mean target class fooling confidence was 0.0008.

5. We have made improvements to the paper to be much more clear about why we care about making attacks that are “realistic” and “disguised”. While past work (e.g., Brown et al., 2017) developed disguised attacks in order to camouflage them to humans, our principal goal is to use feature-level attacks as interpretability tools. Our hypothesis is that feature-level attacks that resemble something other than the target class – i.e. attacks that are “disguised” and “realistic” – can help a human _learn_ about what types of realistic features can be used to influence its predictions. We support this through the design of copy/paste attacks.

---

### Meta-Review · Area_Chair_3xj1 · 2022-08-31

**Recommendation:** Accept
**Confidence:** Certain

**Metareview:**

This paper proposed the use of generative models to create feature level adversarial perturbations.  The resulting attack images have unusual transferability properties.  The reviewers agree that the threat model is interesting, and the clarity and thoroughness of the paper is above the bar.

**Award:**

No

---

### Decision · Program_Chairs · 2022-09-14

Accept